# Impact of bulk-edge coupling on observation of anyonic braiding statistics in quantum Hall interferometers

J. Nakamura [1,2], S. Liang[1,2], G. C. Gardner [2,3] & M. J. Manfra [1,2,3,4,5✉]

Quantum Hall interferometers have been used to probe fractional charge and statistics of quasiparticles. We present measurements of a small Fabry–Perot interferometer in which the electrostatic coupling constants which affect interferometer behavior can be determined experimentally. Near the center of the $\nu = 1/3$ state this device exhibits Aharonov–Bohm interference interrupted by a few discrete phase jumps, and $\Phi_0$ oscillations at higher and lower magnetic fields, consistent with theoretical predictions for detection of anyonic statistics. We estimate the electrostatic parameters $K_I$ and $K_{IL}$ by two methods: using the ratio of oscillation periods in compressible versus incompressible regions, and from finite-bias conductance measurements. We find that the extracted $K_I$ and $K_{IL}$ can account for the deviation of the phase jumps from the theoretical anyonic phase $\theta_a = 2\pi/3$. At integer states, we find that $K_I$ and $K_{IL}$ can account for the Aharonov–Bohm and Coulomb-dominated behavior of different edge states.

[1] Department of Physics and Astronomy, Purdue University, West Lafayette, IN, USA. [2] Birck Nanotechnology Center, Purdue University, West Lafayette, IN, USA. [3] Microsoft Quantum Lab West Lafayette, West Lafayette, IN, USA. [4] School of Electrical and Computer Engineering, Purdue University, West Lafayette, IN, USA. [5] School of Materials Engineering, Purdue University, West Lafayette, IN, USA. ✉email: mmanfra@purdue.edu

Fractional quantum Hall states[1] are predicted to host exotic quasiparticles exhibiting fractional charge and fractional anyonic braiding statistics[2–6]. The fractional charge has been demonstrated through shot noise measurements[7–9], resonant tunneling[10–14], the fractional Josephson relation[15], and interferometry[16–18] for numerous fractional quantum Hall states. Recently, experimental evidence for anyonic statistics was demonstrated through quasiparticle collisions[19] and Fabry–Perot interference[20]. Additionally, interferometry has been used to study the potentially non-Abelian $\nu = 5/2$ state[17,21,22].

Electronic interferometers using quantum point contacts (QPCs) to partition edge states have been proposed as a method to probe both the fractional charge and fractional braiding statistics of quasiparticles[23–26], and substantial theoretical work has been made to understand the behavior of quantum Hall interferometers[27–31], including their application to non-Abelian states[32–36]. For a Fabry–Perot quantum Hall interferometer, the phase difference determining interference will be given by Eq. 1[26,29]:

$$\frac{\theta}{2\pi} = e^* \frac{AB}{\Phi_0} + N_L \frac{\theta_a}{2\pi} \tag{1}$$

$B$ is the magnetic field, $A$ is the area of the interference path set by the gates, $e^*$ is the quasiparticle charge on the interfering edge state normalized to the elementary electron charge $e$, $\Phi_0 \equiv \frac{h}{e}$ is the flux quantum, $N_L$ is the number of localized quasiparticles inside the interferometer, and $\theta_a$ is the anyonic phase associated with the interfering quasiparticles. With QPCs tuned to weak backscattering, oscillations in the conductance across the interferometer will occur with $\delta G \propto \cos(\theta)$, enabling fractional charge and statistics to be probed through transport measurements. For the $\nu = 1/3$ state, $\theta_a$ is predicted to be $\frac{2\pi}{3}$[5,6,37–39], while different anyonic phases are predicted for different fractional quantum Hall states[29,40].

An important consideration for the operation of Fabry–Perot interferometers is the role of Coulomb interactions. It has been shown that the Coulomb interaction between charge in the bulk of the interferometer and charge at the edge has a major effect on interferometer behavior, as it can cause the area of the interferometer to change when charge in the bulk changes, which modifies the Aharonov–Bohm contribution to the interferometer phase[29]; this has important effects in both integer and fractional quantum Hall interference. Strong bulk-edge coupling can result in unusual interference behavior, including a decrease in magnetic flux through the interference path when magnetic field is increased, resulting in positively sloped lines of constant phase in the magnetic field-gate voltage plane. For the integer regime, this has been referred to as the Coulomb-dominated regime, while behavior where the bulk-edge coupling is weak and interference exhibits the conventional negatively-sloped lines of constant phase has been referred to as the Aharonov–Bohm regime. This definition of Aharonov–Bohm and Coulomb-dominated regimes is not as meaningful for fractional states due to the effect of anyonic statistics[41]. Nevertheless, a strong bulk-edge interaction still has critical effects in the fractional regime. Most importantly, the strong bulk-edge coupling can make the anyonic phase unobservable[29], making it important to suppress Coulomb charging effects in interferometers[18,42,43].

In our previous experiment probing the anyonic phase at $\nu = 1/3$[20], the interferometer was in a regime in which the Coulomb charging effects leading to the bulk-edge interaction were highly suppressed, allowing the anyonic phase to be extracted without being significantly reduced. However, this suppression of Coulomb effects also likely results in thermal smearing of quasiparticle transitions in high and low field regions where the bulk becomes compressible.

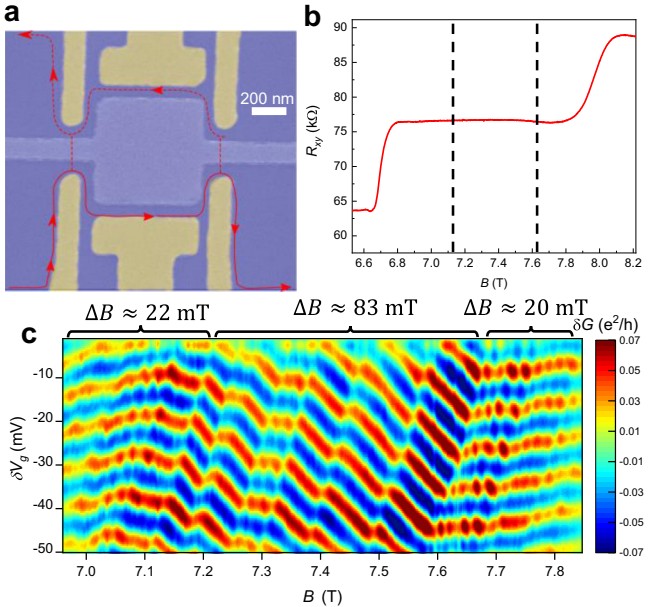

**Fig. 1 Interferometer at the $\nu = 1/3$ fractional quantum Hall state. a** SEM image of interferometer with false color. Yellow regions are the gates that define the interference path. Blue regions correspond to the 2DES. There is a center gate in the middle which is kept grounded and does not affect the 2DES density. Red lines indicate edge state trajectories, with backscattered paths denoted by dashed lines. **b** Bulk $R_{xy}$ measurement showing the $\nu = 1/3$ conductance plateau. The approximate positions in magnetic field where the interferometer transitions from negatively sloped Aharonov–Bohm behavior to flat lines of constant phase with $\Phi_O$ modulations are marked with dashed lines. **c** Interference at $\nu = 1/3$ at a mixing chamber temperature of 10 mK. Near the center there are several discrete jumps in phase which are associated with removal of quasiparticles localized by disorder in the interior of the interferometer. At high and low field the lines of constant phase become nearly independent of magnetic field, but modulations with period approximately $\Phi_O$ can be seen. These modulations are more prominent in the high-field region, particularly close to the transition point at approximately 7.7 T.

Here we report on measurements and analysis of interference in a small gate-defined Fabry–Perot interferometer fabricated on a high-mobility GaAs/AlGaAs two-dimensional electron system (2DES) and specifically designed to investigate the impact of increased bulk-edge coupling. The device, shown in Fig. 1a, has lithographic dimensions of 800 nm × 800 nm, and uses two pairs of gates forming narrow constrictions acting as QPCs to partially backscatter the incoming edge states. A pair of side gates between the two QPCs define the area of the interferometer; the QPC and side gates are highlighted in yellow in Fig. 1a. There is a center gate which is kept at ground and does not affect the 2DES density. The conductance across the device is measured using standard voltage-biased lock-in amplifier techniques. The effective area based on the magnetic field oscillation period at $\nu = 1$ of approximately 21 mT is $\approx 0.2\ \mu m^2$, implying a lateral depletion of approximately 180 nm, similar to previous results for these types of interferometers[18,20].

## Results

A key finding from the Rosenow & Stern model for Fabry–Perot interferometers[44] is that due to the finite energy gap for the creation of quasiparticle and quasihole excitations, there will be a finite range of magnetic field near the center of the state where

filling factor $\nu \equiv \frac{n\Phi_0}{B}$ remains fixed and the bulk is incompressible (here $n$ is the electron density per unit area). In this regime, no quasiparticles are created except for a small number, which are brought to energies significantly smaller than the gap by disorder, and $\theta$ will evolve primarily due to the Aharonov–Bohm phase. Once the magnetic field has been varied sufficiently far from the center of the state that the chemical potential is outside of the spectral gap, quasiparticles or quasiholes should start to enter the device with the expected period $\Phi_0$. At $\nu = 1/3$, the behavior expected for this regime where the bulk is compressible is that upon the addition of magnetic flux $\Phi_0$, a quasiparticle will be removed from the device (or a quasihole will be added), giving a shift in the anyonic phase of $-2\pi/3$ which cancels out the Aharonov–Bohm contribution to the phase. This results in the leading order interference having no magnetic field dependence, but oscillations still occur as a function of gate voltage[29,41]. Higher-order contributions to interference are also expected due to $\Phi_0$ periodic changes in quasiparticle number[44] but are thermally suppressed. In a previous work at $\nu = 1/3$[20], the lines of constant phase were observed to shift from a negative slope near the center to zero slope at high and low field, which is consistent with a shift from constant $\nu$ with an incompressible bulk to constant $n$ with a compressible bulk. $\Phi_0$ modulations from quasiparticle transitions were not observed in the high and low field regions, which was attributed to thermal smearing of the quasiparticle number. Significant thermal smearing is expected due to the small quasiparticle charge $e^* = 1/3$ and large screening needed to suppress bulk-edge coupling[44].

The device we have measured in this work has an area smaller by approximately a factor of 2 compared to the device in ref. [20]. Bulk magnetoresistance $R_{xy}$ is shown in Fig. 1b indicating the $\nu = 1/3$ state and resistance plateau. Conductance data measured across the interferometer as a function of $B$ and gate voltage variation $\delta V_g$ at $\nu = 1/3$ is shown in Fig. 1c ($\delta V_g$ is applied to both side gates, and is relative to $-0.8$ V). A smooth background is subtracted to emphasize the interference oscillations. For this measurement the QPCs were individually tuned to approximately 90% transmission at the center of the $\nu = 1/3$ state to achieve the weak backscattering regime. The overall behavior is similar to[20]: near the center of the plateau the lines of constant phase have a negative slope, which is interrupted by several discrete jumps in phase. At low field and high field the lines of constant phase flatten out, consistent with transitions to a compressible bulk with populations of quasiparticles (at low field) or quasiholes (at high field). Unlike[20], however, there are additional modulations in the interference pattern in the low and high field regions, which have a period of $\approx 22$ mT in the low field region and $\approx 20$ mT in the high field region (Fourier transforms illustrating these periods are shown in Supplemental Fig. 2 and discussed in Supplemental Section 2). This period is close to the Aharonov–Bohm period of $\approx 21$ mT at the integer state $\nu = 1$, indicating that the modulations in the low and high field regions have close to $\Phi_0$ period, as predicted in ref. [44]. When the bulk is compressible, an increase of magnetic flux by $\Phi_0$ will result in a change in quasiparticle number of $-1$, and if the quasiparticles at $\nu = 1/3$ are anyons, this will also result in a jump in phase of $-2\pi/3$, yielding the higher order $\Phi_0$ periodic contributions to inteference. Observations of these $\Phi_0$ oscillations suggests that the reduction in device size has enabled quasiparticle transitions and associated higher-order interference terms to be partially resolved in the compressible regions, giving additional experimental evidence of anyonic statistics. Repeatability of the data is discussed in Supplemental Section 1 and Supplemental Fig. 1.

Interference measurements at elevated temperatures 50 and 90 mK are shown in Fig. 2a, b. At 50 mK the $\Phi_0$ modulations are greatly suppressed, and at 90 mK they are completely washed out,

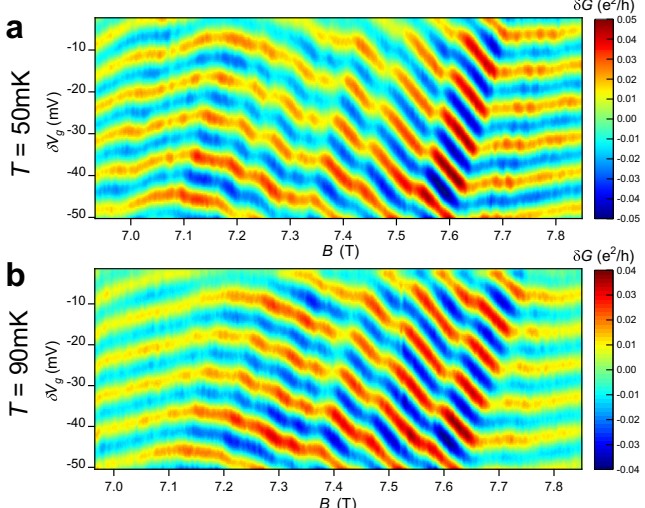

**Fig. 2 Interference at elevated T. a** Interference at $\nu = 1/3$ at a mixing chamber temperature of 50 mK. Interference is still of reasonable amplitude, and a few of the discrete jumps in the center are still visible, but the $\Phi_0$ modulations are only weakly visible. **b** Interference at 90 mK. The $\Phi_0$ modulations are completely washed out, and the interference amplitude in the low and high field regions are greatly reduced.

similar to the observation in ref. [20]. This is consistent with the prediction that the dominant behavior at $\nu = 1/3$ when the bulk is compressible should be oscillations with zero magnetic-field frequency[29,41], while the $\Phi_0$ modulations are a higher-order effect that is more easily thermally smeared. The discrete jumps in the central region also become noticeably less sharp and more smeared out at the higher temperatures.

It is noteworthy in Fig. 1c that the $\Phi_0$ modulations are more prominent in the high-field region than in the low-field region, and we have found that this is usually the case in multiple data sets (see Supplemental Fig. 1). This may suggest an effect of particle–hole asymmetry in terms of confining quasiparticles inside the interferometer, although we do not have a clear explanation for the effect. Though they carry equal and opposite total charge, quasiparticles and quasiholes have different charge distributions. This can be understood from the composite-fermion picture[45] from the fact that quasiparticle states involve addition of charge to excited lambda levels[46]. How the difference in behavior between quasiparticles and quasiholes might contribute to the more clear $\Phi_0$ modulations in the high field region requires further investigation.

A wide range of interferometer behavior beyond the negatively sloped pure Aharonov–Bohm regime has been observed in previous experiments[14,16,18,20,42,47,48]. Theoretical analyses[28,29,44,49] have established that electrostatic interaction parameters are crucial in determining the observed behavior, with key parameters being the edge stiffness $K_I$, which describes the energy cost to vary the area of the interfering edge state, and $K_{IL}$, which parameterizes the coupling of the bulk to the edge. Previous experiments have investigated the case of interference when multiple Landau levels are present and inferred the electrostatic parameters governing interference[43,48]. Device behavior has been modeled by defining an energy function for the electrostatic energy involving these parameters (Eq. 2)[29]:

$$E = \frac{K_I}{2}(\delta n_I)^2 + \frac{K_L}{2}(\delta n_L)^2 + K_{IL}\delta n_L \delta n_I \qquad (2)$$

In this equation $\delta n_L$ is the variation of the charge in the bulk from the background charge (which includes the quantum Hall

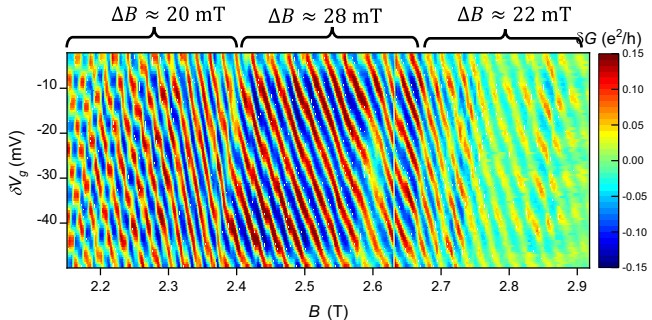

**Fig. 3 Interferometer conductance oscillations versus magnetic field and gate voltage at $\nu = 1$.** Near the center the magnetic field oscillation period is larger, suggesting an incompressible bulk, while at high and low fields the period becomes smaller (while still maintaining an overall negative slope) suggesting a transition to a compressible bulk.

condensate density and the contribution from localized charges), and $\delta n_I$ is the variation in charge at the edge from the ideal value. Minimizing the electrostatic energy will result in variations in the area, $\delta A = -\delta n_L \frac{K_{IL}}{K_I} \frac{\Phi_0}{\Delta \nu B}$. Including this variation in area in Eq. 1 yields Eq. 3[49]:

$$\frac{\theta}{2\pi} = e^* \frac{\overline{A}B}{\Phi_0} - \frac{K_{IL}}{K_I} \frac{e^*}{\Delta \nu} \left( e^* N_L + \nu_{in} \frac{\overline{A}B}{\Phi_0} - \overline{q} \right) + N_L \frac{\theta_a}{2\pi} \quad (3)$$

In Eq. 3 $\overline{A}$ is the average area not including the variations $\delta A$ due to the bulk edge coupling, $\Delta \nu$ is the difference between the filling factor corresponding to the interfering edge state and the filling factor of the next-outer edge state which is fully transmitted (for integer states $\Delta \nu = 1$, while for $\nu = 1/3$ $\Delta \nu = 1/3$). $\nu_{in}$ is the filling factor corresponding to the interfering edge state, and $\overline{q}$ is the background charge (which is primarily determined by the ionized donors and may also be changed by the gate voltage).

Equation 3 implies that in the presence of finite $K_{IL}$, in ranges of magnetic field where the localized quasiparticle number is fixed, the magnetic field oscillation period will be modified from the base value of $\frac{\Phi_0}{e^* \overline{A}}$. In the presence of bulk-edge coupling, this period will change to Eq. 4[41]:

$$\Delta B = \frac{\Phi_0}{e^* \overline{A}} \left( 1 - \frac{K_{IL}}{K_I} \frac{\nu_{in}}{\Delta \nu} \right)^{-1} \quad (4)$$

At $\nu = 1$ we observe a region in magnetic field towards the center of the state where the magnetic field period is larger (Fig. 3), suggesting that in this region $\mu$ is in the gap and the localized electron number is fixed, as predicted by the model in ref. [44]. At higher and lower field the period becomes smaller, suggesting that localized holes and electrons are being added and bringing the period back to the base value of $\frac{\Phi_0}{e^* \overline{A}}$. This allows $\frac{K_{IL}}{K_I}$ to be extracted based on the ratio of the periods, yielding a value of 0.25 (see Supplemental Section 3 and Supplemental Fig. 3). Similar analysis at $\nu = 1/3$ yields $\frac{K_{IL}}{K_I} = 0.24$. These values imply a moderate effect of bulk-edge coupling at $\nu = 1$ and $\nu = 1/3$.

Estimating $\frac{K_{IL}}{K_I}$ from Eq. 4 gives only the ratio of the two parameters, and gives limited insight into the relevant factors which contribute to each term. To extend this analysis and estimate the magnitude of $K_I$ and $K_{IL}$ in our device, we adopt a simple picture in which the energy is a combination of the single-particle energy $E_{sp}$ (which is determined by the external electrostatic confining potential) and the electron interaction energy $E_{int}$ which we estimate by approximating the device as a quantum dot (and extract from zero-field Coulomb blockade measurements). The total energy is a combination of the two terms, $E = E_{sp} + E_{int}$.

In this approximation we estimate the interaction energy by assuming that it can be treated as zero-dimensional quantum dot-like object, so that $E_{int} = \frac{\delta q_{total}^2}{2C}$. $\delta q_{total}$ is the combined bulk and edge excess charge, $\delta q_{total} = e\delta n_I + e\delta n_L$, and $C$ is the electrostatic self capacitance of the device. This yields:

$$E_{int} = \frac{\delta q_{total}^2}{2C} = \frac{e^2}{C} \left( \frac{\delta n_I^2}{2} + \frac{\delta n_L^2}{2} + \delta n_L \delta n_I \right) \quad (5)$$

From this, we can see that the quantum dot-like charging energy $\frac{e^2}{2C}$ contributes to $K_I$, $K_{IL}$, and $K_L$. $\frac{e^2}{2C}$ can be extracted from the height of Coulomb blockade diamonds. For our device the $B = 0$ Coulomb blockade measurements[50] yield $\frac{e^2}{C_{total}} \approx 90\,\mu eV$ (Fig. 4a). This can be refined by subtracting the contribution from the single-particle level spacing due to the finite Density of States (DOS) per unit area $\frac{m^*}{\pi \hbar^2}$ in 2D at $B = 0$ which gives a quantum contribution $\frac{e^2}{C_{quantum}} \approx 18\,\mu eV$ for a device with area $\approx 0.2\,\mu m^2$. Then $\frac{e^2}{C} = \frac{e^2}{C_{total}} - \frac{e^2}{C_{quantum}} \approx 90\,\mu eV - 18\,\mu eV = 72\,\mu eV$.

$E_{sp}$ is set by the confining potential, which increases the system energy when area is changed. This external potential can be approximated as a constant electric field $\mathcal{E}$ assuming the variations in area are small. Then, the additional electrostatic potential the charge added to the edge experiences will be $\mathcal{E}\delta l$, where $\mathcal{E}$ is the electric field and $\delta l = \frac{\delta A}{L}$ is the increase in the radius of the interferometer ($L$ is the perimeter of the interference path). $\delta A$ will depend on the amount of charge added to the edge and the sheet density of the interfering edge state, $\delta A = \frac{\delta n_I}{\rho}$ and $\rho = \frac{\Delta \nu B}{\Phi_0}$. The total change in energy will be the average change in potential times the amount of charge added, $E_{sp} = \frac{e\delta n_I \mathcal{E}\delta l}{2} = \frac{e\delta n_I^2 \mathcal{E}\Phi_0}{2LB\Delta\nu}$. To find the contribution of $E_{sp}$ to $K_I$ we need to determine the value of $\frac{e\mathcal{E}\Phi_0}{LB\Delta\nu}$.

The electric field $\mathcal{E}$ also drives the edge velocity, via $\overrightarrow{v}_{edge} = \frac{\overrightarrow{\mathcal{E}} \times \overrightarrow{B}}{B^2}$, so a measure of velocity can be used to extract the electric field and get $E_{sp}$[26]. For integer states (where $e^* = 1$ and $\Delta \nu = 1$) and weak backscattering, when the experiment of measuring differential conductance as a function of gate voltage or magnetic field and source drain bias $V_{SD}$ is performed, it exhibits a checkerboard pattern with $\delta G \propto \cos\left(\frac{2\pi AB}{\Phi_0}\right) \cos\left(\frac{LeV_{SD}}{2\hbar v_{edge}}\right)$ [18,51–53] (note that this assumes a symmetric potential drop; the symmetry of the potential drop is discussed in Supplemental Section 4). The product of cosines will result in nodes in the oscillation pattern at $\frac{LeV_{SD}}{2\hbar v_{edge}} = \pi(n + 1/2)$, so that the voltage spacing between nodes $\Delta V_{SD} = \frac{2\pi\hbar v_{edge}}{eL} = \frac{hv_{edge}}{eL} = \frac{\Phi_0 \mathcal{E}}{LB}$. This gives $E_{sp} = \frac{e\delta n_I^2}{2}\Delta V_{SD}$.

Differential conductance measurements at $\nu = 1$ are shown in Fig. 4c, and the amplitude versus $V_{SD}$ is plotted in Fig. 4d. The data exhibits the expected checkerboard pattern, and the spacing between minima in the amplitude (corresponding to the nodes in the oscillation pattern) gives $\Delta V_{SD} \approx 162\,\mu V$, and $\frac{e\mathcal{E}\Phi_0}{LB} \approx 162$ $\mu eV$. Combining Eq. 2 with the relationships for $E_{sp}$ and $E_{int}$ gives $K_I = \frac{e^2}{C} + \frac{e\mathcal{E}\Phi_0}{LB} \approx 72\,\mu eV + 162\,\mu eV = 234$ $\mu eV$, while $K_{IL} = \frac{e^2}{C} \approx 72\,\mu eV$. This gives $\frac{K_{IL}}{K_I} = 0.31$. The fact that $\frac{K_{IL}}{K_I} < 0.5$ should place the interferometer in the Aharonov–Bohm regime, which is consistent with the observation of predominantly negatively-sloped behavior at $\nu = 1$ (Fig. 3). Additionally, this value of $\frac{K_{IL}}{K_I}$ is close to the value of 0.25 extracted from Eq. 4, giving further validation for the model.

Strong bulk-edge coupling causes the area of the interference path to decrease when the magnetic field is increased (or when localized quasiparticles are added to the interior of the device), resulting in a positive slope to constant phase lines when $\frac{K_{IL}}{K_I} > 0.5$, which has been observed in some previous

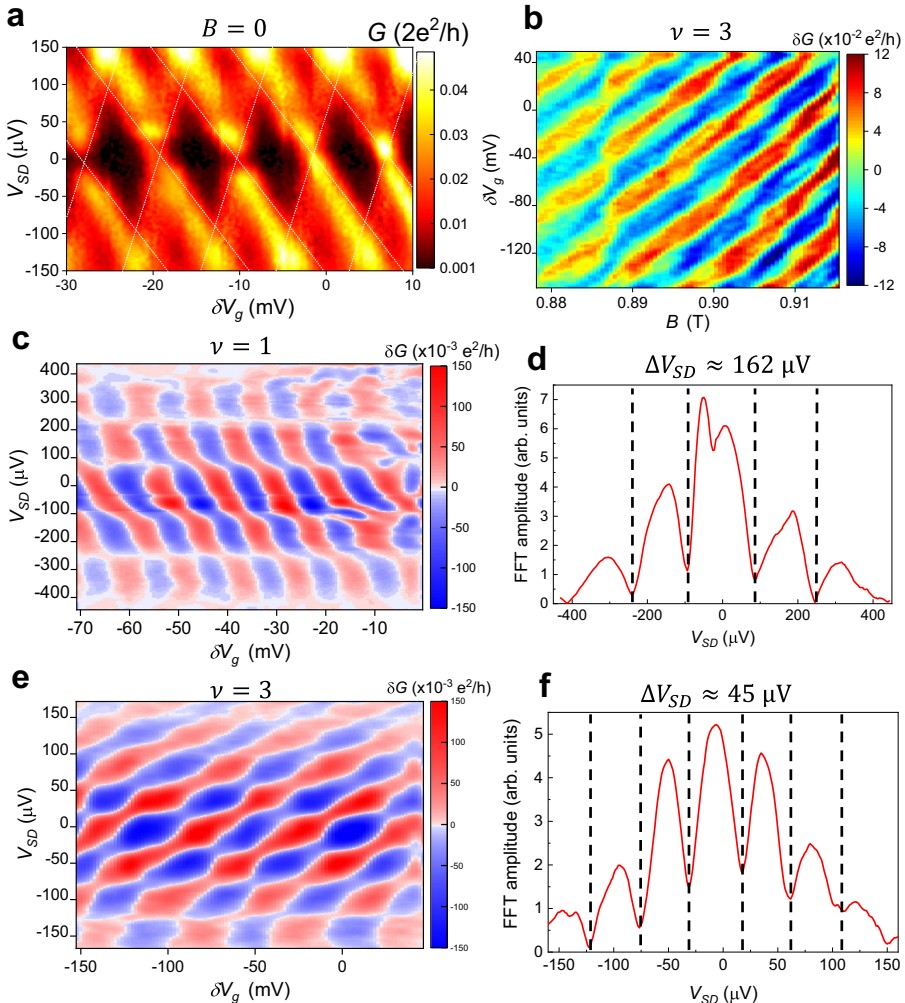

**Fig. 4 Finite-bias measurements. a** Differential conductance measurements at zero magnetic field in the Coulomb blockade regime. The height of the diamond pattern gives the characteristic charging energy $\frac{e^2}{C_{\text{total}}} \approx 90\,\mu\text{eV}$. Note that this energy will include the contribution from the finite DOS, which should be subtracted to yield the electrostatic component that contributes the $K_{IL}$. This yields $\frac{e^2}{2C} \approx 72\,\mu\text{eV}$. **b** Interference data for the innermost edge mode at $\nu = 3$. The QPCs are tuned to partially reflect the inner mode and fully transmit the two outer modes. The positive slope indicates that under these conditions the device is in the Coulomb-dominated regime where the bulk-edge interaction is strong. **c** Differential conductance measurements at $B = 2.5$ T, the $\nu = 1$ state. The checkerboard pattern suggests that the bias is close to symmetric, although the fact that there is some tilt to the pattern suggests that there is some asymmetry. **d** Oscillation amplitude from a FFT versus $V_{SD}$ for $\nu = 1$. The minima in the pattern correspond to the nodes in the checkerboard pattern, allowing extraction of $\Delta V_{SD} \approx 162\,\mu\text{V}$. **e** Differential conductance measurement for the inner mode at $\nu = 3$. **f** FFT amplitude versus $V_{SD}$. The spacing between minima gives $\Delta V_{SD} = 45\,\mu\text{V}$, indicating a relatively low velocity which is to be expected for an inner mode.

experiments[14,16,42,47]. With $B$ set to $\nu = 3$, interference data is shown for the innermost mode in Fig. 4b, where the QPCs are tuned to weakly backscatter the innermost mode and fully transmit the outer modes. There is an overall positive slope to the data, indicating that unlike at $\nu = 1$, the device is in the Coulomb-dominated regime and the bulk-edge interaction is strong. This is also supported by the fact that the magnetic field oscillation period is approximately $\frac{\Phi_0}{2}$, similar to previous experiments[16,42] and theory[28,29] for Coulomb-dominated interference of an inner mode at $\nu = 3$.

Differential conductance measurements at $\nu = 3$ are shown in Fig. 4e, and the oscillation amplitude versus $V_{SD}$ shown in Fig. 4f. From this data $\Delta V_{SD} = 45\,\mu\text{V}$, implying a lower velocity and smaller edge stiffness than at $\nu = 1$. This low velocity can be understood from the fact that an inner edge state is being interfered; the inner edge state will be positioned at a region with a more shallow confining potential, resulting in a lower electric field and thus lower velocity[18,54,55], making it easier for the area

enclosed by this edge state to change. The measured value yields $K_I = \frac{e^2}{C} + e\Delta V_{SD} = 72\,\mu\text{eV} + 45\,\mu\text{eV} = 117\,\mu\text{eV}$ and $K_{IL} = \frac{e^2}{C} \approx 72\,\mu\text{eV}$. This gives $\frac{K_{IL}}{K_I} = 0.62$. Since $\frac{K_{IL}}{K_I} > 0.5$, the device is predicted to be in the Coulomb-dominated regime, consistent with the observed interference behavior in Fig. 4b. This gives additional validation for this method of calculating $K_I$ and $K_{IL}$ since this model is able to correctly predict Coulomb-dominated behavior. Additionally, because this is an inner edge mode, it may enclose an area smaller than the overall interferometer area, and thus might have greater bulk-edge coupling and $K_{IL}$ than estimated from zero-field Coulomb blockade. This would drive the device further into the Coulomb-dominated regime.

On the other hand, the outer modes at $\nu = 3$, corresponding to the two spin configurations of the $N = 0$ Landau level, exhibit negatively sloped behavior, indicating that a higher velocity due to a steeper confining potential farther out at the edge gives a stronger $K_I$ (see Supplemental Fig. 5). The outermost edge also

exhibits the period-halving effect which has been observed previously[56–58] and attributed to inter-edge interaction[59] or electron pairing[60].

Recent experiments have extended quantum Hall interferometry to graphene[52,53], and provide another opportunity to apply this method for using $K_I$ and $K_{IL}$ to understand Aharonov–Bohm versus Coulomb dominated behavior. In ref. [52] the smallest device with area 3.1 μm² had estimated charging energy $E_c = 18$ μeV and $\Delta V_{SD} \approx 70$ μV from differential conductance measurements when interfering the outer edge state at $\nu = 2$. This yields $\frac{K_{IL}}{K_I} \approx \frac{18\,\mu V}{18\,\mu V + 70\,\mu V} \approx 0.2$, placing the device in the Aharonov–Bohm regime, which is consistent with the negatively sloped lines of constant phase observed. Similarly, the 3 μm² device in ref. [53] had an estimated $E_c = 16$ μeV and $\Delta V_{SD} = 50$ μV for the inner mode at $\nu = 2$, yielding $\frac{K_{IL}}{K_I} \approx 0.24$, also in concordance with the observed negatively-sloped Aharonov–Bohm behavior. This suggests that the method for analyzing $K_I$ and $K_{IL}$ can also be applied to graphene devices, which are promising for probing exotic statistics.

The finite-bias behavior of interference at the $\nu = 1/3$ state is expected to be modified by Luttinger liquid effects[26,61]. Reference [26] analyzes the current through interferometers as a function of $V_{SD}$ and $T$, and finds that while integer states should have uniform spacing of nodes (as discussed in the previous section), for fractional states the innermost nodes will have a narrower spacing than the outer nodes. At low temperature, the positions of the nodes will approximately be given by Eq. [6]:

$$eV_{SD} = \frac{h\nu_{edge}}{e^* L}\left(n + \frac{1+g}{2}\right), n = 0, 1, 2, \ldots \quad (6)$$

Here $g$ is the tunneling exponent, expected to be 1/3 for the $\nu = 1/3$ state. This implies that the innermost nodes will have a narrower spacing than the outer ones, which will have a spacing $e\Delta V_{SD} = \frac{h\nu_{edge}}{e^* L}$. At high temperatures, ref. [26] predicts that the innermost nodes will move outward and reach the same spacing as the outer nodes, so that node spacing is uniform, as in the integer case. Since this theory calculates the total current, it is most convenient to work with the DC current oscillation amplitude $\delta I$ rather than the differential conductance; therefore, we have measured both $\delta G$ and $\delta I$ as a function of $V_{SD}$ at $\nu = 1/3$.

The differential conductance for $\nu = 1/3$ is plotted in Fig. 5a, with the amplitude shown in b. The central separation between the innermost nodes is $\approx 120$ μV, while the separation between the outer nodes is $\approx 190$ μV. More direct comparison to[26] can be made by measurements of oscillations in the DC current, shown in Fig. 5c, d. The outer nodes have a separation of $\approx 197$ μV, while the inner ones have a separation of $\approx 167$ μV. This is consistent with the expectation that the inner nodes should have a narrower spacing; however, the ratio of the inner to the outer node spacing is 0.85, which is somewhat larger than the value of 2/3 predicted by Eq. [6] from[26]. A possible explanation for this discrepancy is that the large biases applied in these measurements cause significant heating of the electrons in the device, shifting behavior towards the high-temperature limit of uniform node spacing. Additionally, at elevated mixing chamber temperature the innermost node moves to higher $V_{SD}$, approaching the spacing of the outer nodes as anticipated by[26]; see Supplemental Fig. 6 and Supplemental Section 5. The qualitative agreement suggests that this theory can be used to extract $\nu_{edge}$ from the outer node spacing of $\Delta V_{SD} \approx 197$ μV (the differential conductance measurement shows similar values for the outer node spacing). Then, $E_{sp} = \frac{e\delta n_I^2 \mathcal{E} \Phi_0}{2LB\Delta\nu} = e\Delta V_{SD}\frac{e^* \delta n_I^2}{2\Delta\nu}$; with $e^* = 1/3$ and $\Delta\nu = 1/3$ for the $\nu = 1/3$ state, $E_{sp} = \frac{e\delta n_I^2}{2}\Delta V_{SD}$ as for the integer case. Using $\Delta V_{SD} \approx 197$ μV gives $K_I \approx 72$ μeV + 197 μeV = 269 μeV, while as

before $K_{IL} = \frac{e^2}{C} \approx 72$ μeV. This gives $\frac{K_{IL}}{K_I} = 0.27$, close to the value of 0.24 extracted from Eq. [4]. A possible limitation to this approach is that the equations for current in ref. [26] were developed without including bulk-edge coupling. Future theoretical work could refine this analysis by analyzing bulk-edge coupling corrections to the positions of nodes and improve the accuracy of $K_I$.

The narrower spacing of the innermost nodes, being a signature of Luttinger-liquid behavior, contrasts with the nearly uniform node spacing observed for the integer states at $\nu = 1$ and $\nu = 3$ in Fig. 4a, c, which is expected for Fermi liquids. Previous experimental evidence for Luttinger-liquid behavior of fractional quantum Hall edge states has been seen in tunneling experiments[62,63], while here we have shown evidence through interferometry.

Several discrete phase jumps can be seen in Fig. 1a, similar to previous observations[20], which may be caused by the anyonic phase when the number of localized quasiparticles inside the interferometer changes. To extract the values of these phase jumps, we have calculated the phase at each value of the magnetic field by taking Fourier transforms along cuts parallel to the lines corresponding to discrete jumps. Then we subtract off the Aharonov–Bohm contribution to the phase (which simply results in continuous phase evolution and a constant linear slope in phase vs. $B$). The process for extracting the phases in this way is discussed in detail in the Supplemental Section 5 and illustrated in Supplemental Fig. 7; this method should enable a more accurate phase extraction than the fitting method in[20] and has the additional advantage of not needing the position of each jump to be specified. The resulting phase after subtracting the Aharonov–Bohm contribution should be due to the anyonic contribtuion, and is plotted in Fig. 6a. As can be seen in Fig. 1a, some of the phase jumps are very close to each other, so that the individual jumps in phase are not readily resolvable; in particular there appear to be two very close jumps at $\approx 7.28$ T and three close jumps at $\approx 7.37$ T. While the individual phase jumps cannot be isolated, the combined phase jump can be extracted from the data. At low fields (below approximately 7.2 T) and high fields (above approximately 7.7 T) the phase exhibits a staircase pattern due to $\approx \Phi_0$ periodic additions of quasiparticles, although since there is still significant smearing, this staircase pattern is not sharp.

The values of the phase jumps (both the individual ones from isolated jumps and the combined ones when multiple are very close) are listed in Fig. 6a, and the corresponding part of the data where the jumps occur is indicated in (b). These values are calculated by taking the average value of the phase on each plateau and subtracting the adjacent values to get the jump in phase. Averaging all the jumps (and taking into account the fact that some of the changes in phase are most likely due to multiple discrete jumps) yields an average change in phase $\frac{\Delta\theta}{2\pi} = -0.24 \pm 0.04$ (uncertainty is estimated from the standard deviation of the phase jumps). The magnitude of the observed phase jumps is smaller than the value of $\Delta\theta = -\theta_a = -\frac{2\pi}{3}$ expected from theory (here the negative sign occurs because $\Delta\theta$ is for a change in quasiparticle number by $-1$, and the phase is defined from $-\pi$ to $+\pi$). However, theoretical analyses[29,41,49] predict a modification to the value of the phase jump that occurs when the quasiparticle number changes by $-1$ due to bulk edge coupling (Eq. [7]):

$$\frac{\Delta\theta}{2\pi} = -\frac{\theta_a}{2\pi} + \frac{K_{IL}}{K_I}\frac{e^{*2}}{\Delta\nu} \quad (7)$$

This modification comes about because when a quasiparticle enters the bulk, its electric charge will cause the area of the interferometer to change, leading to a change in the

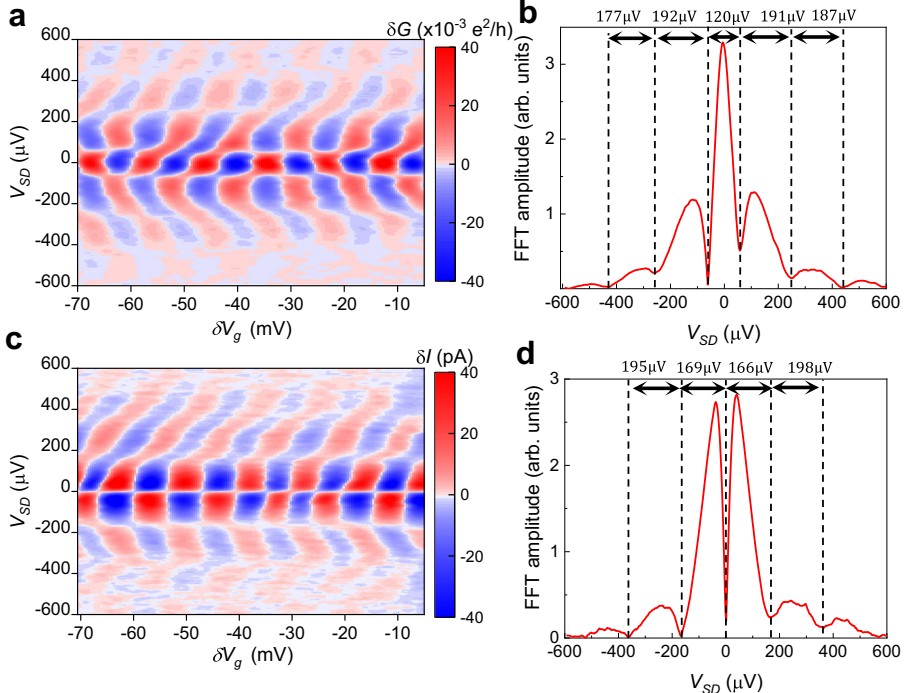

**Fig. 5 Finite-bias measurements at $\nu = 1/3$. a** Differential conductance measurements at $\nu = 1/3$, $B = 7.4$ T. **b** Differential conductance oscillation amplitude vs. $V_{SD}$. The spacing between nodes (which appear as minima in the plot) are indicated. It is noteworthy that the spacing between the central nodes is somewhat more narrow than the outer nodes, which may be an indication of Luttinger liquid behavior. **c** DC current versus $\delta V_g$ and $V_{SD}$. **d** DC current oscillation amplitude versus $V_{SD}$. The minima in the amplitude correspond to nodes in the interference pattern. As predicted by[26], the central nodes have narrower spacing than the outer ones.

Aharonov–Bohm phase in addition to the anyonic phase. This correction can be included into the extraction of the anyonic phase at $\nu = 1/3$ by $\frac{\theta_a}{2\pi} = -\overline{\frac{\Delta\theta}{2\pi}} + \frac{1}{3}\frac{K_{IL}}{K_I}$. Using the value of $\frac{K_{IL}}{K_I} = 0.24$ extracted from Eq. 4 gives $\frac{\theta_a}{2\pi} = 0.32 \pm 0.05$, while using $\frac{K_{IL}}{K_I} = 0.27$ from finite bias measurements yields $\frac{\theta_a}{2\pi} = 0.33 \pm 0.05$; these values are close to the value of $\theta_a = \frac{2\pi}{3}$ from theoretical and numerical studies[6,37–39]. Thus, although the bulk-edge interaction reduces the observed phase jumps, accounting for this effect indicates that the anyonic phase is close to the theoretically predicted value, giving strong support to the theoretical works[29,41,49] and consistent with previous experiments at $\nu = 1/3$[19,20].

An additional effect of bulk-edge coupling in the integer quantum Hall regime is discrete jumps in phase with when the localized electron number changes[29,43,48]; at $\nu = 1$ in our data, jumps occur consistent with this in some ranges of magnetic field (see Supplementary Section 7 and Supplementary Fig 8), and similar effects can be seen at $\nu = 3$ in Fig. 4b where bulk-edge coupling is strong. It should be considered whether the discrete jumps in phase observed at $\nu = 1/3$ can be explained by bulk-edge coupling alone rather than anyonic statistics[41].

From Eq. 3 and ref. [49], the change in phase for removing a quasiparticle is $\frac{K_{IL}}{K_I}\frac{e^{*2}}{\Delta\nu}$ if $\theta_a$ is assumed to be zero. For $\nu = 1/3$ where $\Delta\nu = 1/3$ and $e^* = 1/3$, this would result in phase jumps of $\frac{\Delta\theta}{2\pi} = \frac{1}{3}\frac{K_{IL}}{K_I}$. Using the value of $\frac{K_{IL}}{K_I}$ of 0.27 extracted from the period measurements (with the value from differential conductance measurements being similar, assuming that the interpretation at $\nu = 1/3$ is correct) gives an expected $\frac{\Delta\theta}{2\pi} = 0.09$. This value is significantly different from the phase jumps observed in the data in Fig. 6, and is of opposite sign. This suggests that while bulk-edge coupling does reduce the value of the phase jumps observed in this device, the phase jumps cannot be explained by bulk edge

coupling alone without anyonic statistics. An assumption made in this analysis is that the charge of the localized quasiparticles is equal to the theoretically predicted value $e^* = 1/3$, whereas a larger localized charge would result in a greater phase jump contribution from bulk-edge coupling. Scanning probe experiments have observed $e/3$ localized charge at the $\nu = 1/3$ state[64], supporting the assumption of fractional charge. Additionally, in previous measurements of a larger device with weak bulk-edge coupling discrete phase jumps close to the expected anyonic phase of $\frac{2\pi}{3}$ were observed[20], slightly larger than the jumps measured in the present device. If the phase jumps were caused only by bulk edge coupling they would be expected to be significantly larger in the smaller device with greater $K_{IL}$. The fact that they are instead slightly smaller is consistent with the phase jumps being reduced by bulk-edge coupling, but not with being caused exclusively by it.

In conclusion, we have demonstrated experimental evidence for multiple theoretical predictions of quantum Hall interferometers. We have observed $\Phi_0$ period modulations in interference at the $\nu = 1/3$ state, which are a signature of anyonic statistics when the bulk is compressible. We have demonstrated two approaches for estimating the impact of bulk edge coupling: using the ratio of the magnetic field periods, and extracting the electrostatic coupling constants $K_I$ and $K_{IL}$ directly from finite bias measurements. Uneven node spacing observed at $\nu = 1/3$ in finite-bias measurements indicates Luttinger liquid behavior. Although our model makes several simplifications, we find that this approach validates theoretical predictions for distinguishing between the Aharonov–Bohm and Coulomb-dominated regimes in the integer quantum Hall regime. Accounting for the correction to $\Delta\theta$ from finite $\frac{K_{IL}}{K_I}$ yields values of $\theta_a$ in agreement with the theoretically predicted value at $\nu = 1/3$, supporting previous experiments. An important finding is that the parameter $\frac{K_{IL}}{K_I}$ can

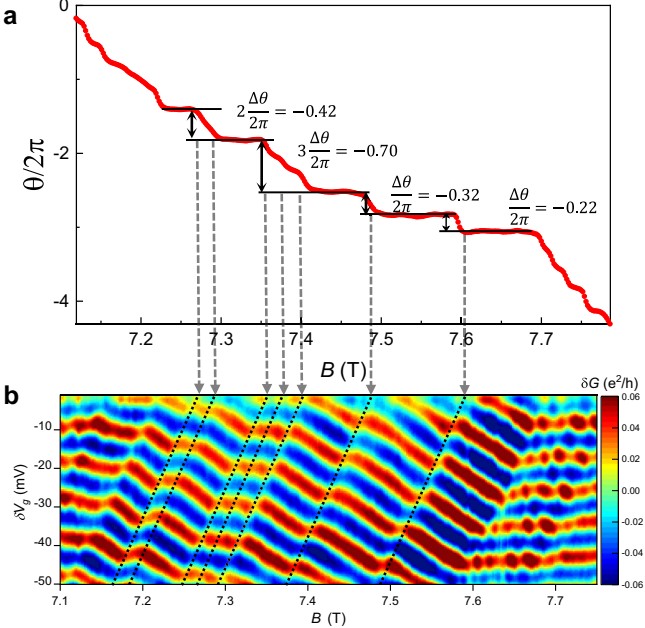

**Fig. 6 Phase versus magnetic field. a** Phases extracted from Fourier transforms of the data in Fig. 1a. The FFTs are performed along cuts of the conductance parallel to discrete phase jumps. The phase is evaluated at the peak frequency which corresponds to the Aharonov–Bohm oscillation frequency. The Aharonov–Bohm effect gives a constant linear change of phase with B which has been subtracted off to yield the contribution from localized quasiparticles, and plateaus in phase occur which correspond to the regions between phase jumps. The change in phase for the discrete jumps are indicated; the leftmost discrete jump appears to correspond to two closeby jumps, while the second from the left appears to consist of three closeby jumps; in these cases, the individual phase jumps are not readily resolved, but the total phase change can be calculated and divided into average individual phase changes. **b** Raw data (repeated from Fig. 1a) indicating where the discrete jumps in (**a**) occur in the data.

vary between different edge states in the same device, which makes inner edge states more likely to be Coulomb-dominated. This work will inform future experimental and theoretical analysis of quantum Hall interferometry.

## Methods

This interferometer utilizes a high mobility GaAs/AlGaAs heterostructure grown by molecular beam epitaxy[65,66]. The bulk electron density is approximately $0.6 \times 10^{11}$ cm$^{-2}$ and mobility is $3.2 \times 10^{6}$ cm$^2$ V$^{-1}$s$^{-1}$. The structure also includes additional screening wells with a setback of 25 nm from the main quantum well to reduce the charging energy and bulk-edge coupling so that anyonic statistics can be observed. The screening well design may also enhance the steepness of the confining potential[18,67], which may be important for preventing edge reconstruction that may lead to dephasing by neutral modes[68–70]. There are metal gates around the Ohmic contacts on the surface and back side of the chip which are negatively biased to deplete electrons from the screening wells so that transport is measured only in the primary quantum well. Though the structure has the same layer stack as the one in ref. [20], the wafer is different and was grown at a different time.

Optical lithography and wet etching were used to define the mesa. Ni/Au/Ge Ohmic contacts were deposited and annealed to make electrical contact to the 2DES. Electron beam lithography and electron beam evaporation (5 nm Ti/10 nm Au) were used to define the interferometer gates. Optical lithography and electron beam evaporation (20 nm Ti/150 nm Au) were used to define bondpads and the surface gates around the Ohmic contacts. The substrate was mechanically polished to make it thin enough to define metal backgates to deplete electrons in the bottom screening well, which were patterned by optical lithography and deposited by electron beam evaporation (100 nm Ti/150 nm Au).

Measurements are performed using standard voltage-biased low frequency lock-in amplifier techniques with a typical excitation of 5 μV and frequency of 37 Hz in a dilution refrigerator with a base mixing chamber temperature of 10 mK.

## Data availability
The transport data generated in this study have been deposited in the Zenodo database under accession code https://doi.org/10.5281/zenodo.5750332. The data is also provided with this paper.

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

## Acknowledgements

This work is supported by the U.S. Department of Energy, Office of Science, Office of Basic Energy Sciences, under award number DE-SC0020138; M.M., J.N. and S.L. were supported by this grant. G.G. acknowledges support from Microsoft Quantum. The authors thank Bertrand Halperin for valuable suggestions to improve the manuscript. M. J. Manfra thanks Dima Feldman and Steve Simon for fruitful discussions during the early phases of this project.

## Author contributions

J.N. and M.M. designed the heterostructures and experiments. S.L. and G.G. conducted molecular beam epitaxy growth. J.N. fabricated the devices, performed the measurements, and analyzed the data with input from M.M. J.N., and M.M wrote the manuscript with input from all authors.

## Competing interests

The authors declare no competing interests.
