## [Peer Review File · Nature Communications]

Reviewer #1 (Remarks to the Author):

This paper studies the different behavior of the quantum Hall interferometers on GaAs at different filling factor ν . At $\nu = 1/3$, the Φ_0 period in interference at higher and lower magnetic field attributed to the higher-order interference terms is observed, which is considered as the evidence of anyonic statistics. And the paper utilizes two different methods to estimate the bulk-edge coupling strength, which is quantified by K_{IL}/K_I and the results are consistent. In addition, the correction to the measured size of the discrete jumps in anyonic phase at $\nu = 1/3$ is well explained by including the effect of bulk-edge coupling. In general, I think this is a nice work. This work systematically studies the influence of bulk-edge interaction on the performance of quantum Hall interference. The presented results and method are meaningful for further research on anyonic braiding and the analysis of quantum Hall interferometry. I would like to recommend this work to be published.

However, I have some questions below:

1. According to page 2, when in the finite range of the magnetic field near the center of the state where filling factor remains fixed, no quasiparticles can be created. However, the phase jump caused by the anyonic phase occurs in the same regime. How to explain this contradiction?

2. I think it's better to introduce the structure of device (which one is the QPC and gate in FIG.1) and measurement setup more detail before present the measurement results.

3. The connection between higher-order interference term and anyonic statistics is not well explained in page 3, which makes the conclusion seem abrupt.

4. At $\nu = 1/3$, the spacing between nodes ΔV_{SD} at finite-bias current measurement is used to calculate K_I , why not use the data from differential conductance, since theory of DC current in the ref [25] doesn't include bulk-edge coupling? And in page 6, the ratio of the inner to the outer node spacing 0.85 is larger than $2/3$, not smaller.

5. The article lacks commas in some places. In page 2, when introduce the filling factor, n is not well defined. It's better to unify the position of square index like $\frac{K_I}{2} (\delta n_I)^2 + \frac{K_I}{2} (\delta n_L)^2$ in

Eqn.2. In page 5, single-particle energy should be $\frac{E_{SD}}{2\pi} = \frac{e}{3} \frac{\delta n_I^2}{K_L} \Delta V_{SD}$, which lack the "e" in the main text. In FIG.5, it's better to unify the form of the axis and legend for 5a and 5c. In page

7, according to Eqn.7, $\theta = \overline{\Delta\theta} + \frac{1}{K_{IL}}$ not $\theta = -\overline{\Delta\theta} - \frac{1}{K_{IL}}$.

Reviewer #2 (Remarks to the Author):

Referee report on the manuscript NCOMMS-21-25268-T by James Nakamura, S. Liang, Geoffrey Gardner, and Michael Manfra

In their manuscript, the authors study the magnitude of the anyonic exchange phase in the fractional quantum Hall state at filling factor $1/3$ by analyzing the magnitude of

so-called phase jumps due to a change of quasi-particle number inside a Fabry-Perot interferometer, and find very good agreement with the theoretically expected value. While the same quantity was studied earlier by the same authors in their reference [19] and also by an alternative method in Ref. [18], I believe that the authors have made significant progress in the present manuscript, which clearly justifies publication in Nature Communications. In particular, in the present manuscript the authors present an in-depth study of Coulomb corrections to the anyonic exchange phase, which were not studied in detail in [19] (which was less important in that reference due to a different device geometry). In order to quantitatively study Coulomb corrections to the exchange phase, the authors use a smaller device with increased Coulomb interaction in the present manuscript. In addition, the authors verify their interpretation of the experimental data by considering additional observables not studied earlier, in this way

adding credibility to their theoretical modelling. I warmly recommend the well written manuscript for publication in Nature Communications.

The magnitude of Coulomb corrections to the exchange phase is closely related to the magnitude of the Coulomb interaction between edge and bulk of the interferometer, divided by the charging energy of electrons travelling along the edge of the interferometer. The authors present a very thorough analysis, measuring the ratio of coupling strengths in several different ways: i) they determine the charging energy of the device at zero magnetic field, and measure the quantum Hall edge velocity by analyzing the checkerboard pattern of the non-linear voltage dependence of the interference signal, for both integer and fractional states. In addition, ii) they determine the ratio of bulk-edge coupling and charging energy by looking at the ratio of magnetic field periods in the regimes of constant chemical potential and constant density, iii) compare this ratio between integer and fractional fillings, thus demonstrating consistency of their results. As yet another consistency check, iv) the authors demonstrate that their determination of Coulomb corrections is consistent with the transition between Aharonov-Bohm interference for the two outermost edge modes and Coulomb dominated interference for the innermost edge modes at integer filling 3.

In addition, the authors demonstrate consistency of the nonlinear interference pattern at fractional filling $1/3$ with the predictions of Luttinger liquid theory. They also use an improved scheme for determining the magnitude of phase jumps.

In summary, the authors not only strengthen the case for the observation of the anyonic exchange phase at fractional filling, but also study a wealth of interesting physical phenomena, contributing to a deeper understanding of fractional quantum Hall interferometers. The research is performed very thoroughly, and the manuscript is written with meticulous attention to detail.

Reviewer #3 (Remarks to the Author):

The manuscript by Nakamura and co-workers reports electronic transport measurements in a Fabry-Perot interferometer in the fractional quantum Hall regime. This is an extension, but with significant progress and additions, of their previous papers where they reported discrete phase jumps in Aharonov-Bohm interference of fractional charges, which they ascribed to anyonic phase arising from changes in the number of quasiparticles in the interferometer. In this work, by using a smaller device the authors examine the impacts of Coulomb interactions on the interference pattern and phase jumps predicted by theory. They succeed in observing the predicted flux-periodic oscillations in the compressible regime, which were not visible in the previous study due to thermal smearing. Using various methods, the authors deduce the important parameters describing the bulk-edge Coulomb coupling, which allowed them to discuss quantitatively the impacts of bulk-edge coupling on the phase jumps on adding/removing quasiparticles.

This is a beautiful work reporting amazingly rich and important data. It is surprising that such systematic experimental data are provided allowing for a stringent test of the theory in every detail. I have no doubt that the manuscript should be published. Yet, as the topic is so intricate, I have several suggestions and questions as follows which the authors may want to consider for clarification before publication.

1) The most important questions are whether or not the observed discrete phase jumps are indeed due to anyonic phase and the observed slightly reduced magnitude of the phase jumps are due to bulk-edge coupling, as discussed by the authors in the last section. I believe that the authors' analysis and interpretations are correct, but complications arising from the sign conventions make it rather troublesome to confirm this. For example, the authors observe phase jumps of $\Delta\theta/2\pi = -0.24$, with the negative sign explained as due to the fact that quasiparticle number decreases upon increasing magnetic field. However, eq. (20) of ref. 28, which is given for the case of quasiparticle number changing by -1 , shows that the phase jump (normalized by 2π) varies

from $2/3$ for the weak coupling limit ($K_{IL}/K_I = 0$) to 1 for the strong coupling limit ($K_{IL}/K_I = 1$). Since the phase is modulo 2π , this is equivalent to varying from $-1/3$ (weak-coupling limit) to 0 (strong-coupling limit), consistent with the authors' observation. But, in this comparison, the negative sign is just a matter of choice in the domain of definition, rather than being due to the fact that the quasiparticles are being removed. Perhaps the confusion occurs because the authors first introduce eq. (7) in the form for the case of adding a quasiparticle and then try to use it for comparison with experiments where they define the phase jump for removing quasiparticles. I suggest the authors always use equations for removing quasiparticles (or decreasing the net quasiparticle number) as in ref. 28 and state that they define the phase in the range from $-\pi$ to π .

2) On page 7, right column, there must be something wrong with the equation, $\theta_a/2\pi = -\overline{\Delta\theta}/2\pi - (1/3)K_{IL}/K_I$. If we plug in $\overline{\Delta\theta}/2\pi = -0.24$ and $K_{IL}/K_I = 0.24$, we have $\theta_a/2\pi = -(-0.24) - (1/3)*0.24 = 0.16$ instead of 0.32. While removing the first minus sign on the right-hand side gives a correct result (0.32), the minus sign in the second term is inconsistent with eq. (7). It is not clear whether this is due to the difference between adding and removing a quasiparticle.

3) For integer QH states the phase jumps associated with a change of N_L by -1 equals K_{IL}/K_I (eq. 19 of ref. 28). As the authors note at the end of page 7, the bulk-edge coupling can cause discrete phase jumps even for electrons and, if so, the ratio K_{IL}/K_I would be extracted directly from the phase jumps, as has been done in Ref. 42. However, in the $\nu = 1$ data shown in Fig.3, such discrete phase jumps are not observed, even when localized electrons or holes are introduced at lower and higher fields as manifested by the changes in the magnetic-field period. (Instead, we see modulation on the stripes, indicating contribution from other Fourier components.) I would like the authors to elaborate a bit more on the behavior of integer systems in their sample in this respect, including the $\nu = 3$ data in Fig. 4 (where features like phase jumps are seen).

4) The authors consistently use the value of $K_{IL} = 72$ ueV derived from the Coulomb diamonds at zero field for $\nu = 3$ as well as for $\nu = 1$ and $1/3$. As the $\nu = 3$ region is surrounded by a relatively wide $\nu = 2$ incompressible strip, its island size could be smaller than for the case of $\nu = 1$ and $1/3$. Could this affect the value of K_{IL} and the authors' discussions on the $\nu = 3$ data?

5) At three places in the manuscript, the authors use the verb "obscure" to describe the impact the bulk-edge coupling has on the anyonic phase. I doubt whether "obscure" is the right word because it sounds as if bulk-edge coupling only makes the anyonic phase difficult to observe but does not affect the magnitude of the phase jump.

Minor or technical points:

- page 1

Φ_0 in the abstract is not defined.

- page 1, left column

The sentence in the introduction, “Additionally, evidence of non-Abelian braiding statistics has been seen in interferometry experiments at $\nu = 5/2$,” might be misleading. It sounds as if the non-Abelian braiding statistics of the $5/2$ state have already been experimentally demonstrated.

- page 1, left column

About Eq. (1) and wherever relevant, the authors should mention that the charges, when expressed with an asterisk, denote charges normalized by the elementary charge.

- page 1, right column

The sentence in the introduction, “Strong bulk-edge coupling can result in unusual interference behavior, including a decrease in magnetic flux through the interference path when magnetic field is increased, resulting in positively sloped lines of constant phase.”

The authors should mention that this is referring to a plot in the magnetic field-gate voltage plane; otherwise, the term “positively sloped” would not make sense.

- page 2, left column

The statement about the constant-filling regime, “in this regime ν quasiparticles are created and θ will evolve primarily due to the Aharonov-Bohm phase” appears a bit confusing, because in the experimental data shown in Fig. 1c several phase jumps due to the creation of quasiparticles are observed in the constant-filling regime.

- page 2, left column

Incomplete sentence: "At $\nu = 1/3$ the behavior expected for this regime where the bulk is compressible that is upon the addition of magnetic flux Φ_0 a quasiparticle will be removed from the device (or a quasihole will be added)..."

- page 2, left column

The quantity "n" should be defined when it first appears in the equation $\nu \equiv n\Phi_0/B$. This is important, since otherwise "constant n" (page 2, right column) would not make sense.

- page 2, left column

due Φ_0 periodic - due to Φ_0 periodic

- page 2, left column

A conjunctive is missing in the long sentence starting with "In a previous work at $\nu = 1/3$ [19] the lines of constant phase were observed to shift from a negative slope near the center to zero slope at high and low field, ..."

- page 3, right column

"with zero magnetic field frequency" - "with zero magnetic-field frequency"

- page 6, right column

"More direct comparison to [25] can be made by to measurements of oscillations in the DC current"

- "More direct comparison to [25] can be made by measurements of oscillations in the DC current"

- page 6, right column

"which is somewhat smaller than the value of $2/3$ predicted by Eqn. 6"

Correctly, it should read, "which is somewhat larger than the value of $2/3$ predicted by Eqn. 6."

- page 7, right column

"note however that the difference in sign is accounted for the fact that ..."

 "note however that the difference in sign is accounted for by the fact that ..."

- page 9, left column

A predicate verb is missing in "Optical lithography and wet etching to define the mesa."

- page 9, left column

"mechanically polished to to make"  "mechanically polished to make"

- In Fig. 4c-4f and Fig. 5 (and many of the figures in the supplementary material), the fonts of the labels on ticks, axes, and color scale bars are too small.

- Reference 40 is already published.

Rep. Prog. Phys. 84, 076501 (2021).

We are pleased to learn that all referees recommend our manuscript for publication with revisions. We thank all the reviewers for their constructive comments and suggestions, and we have modified our manuscript according to their feedback. We hope that our modifications in both the main text and the supplemental materials have improved our manuscript sufficiently to move forward with publication.

Below, we address the specific questions from each of the reviewers, and we have also included a list of the changes made to the manuscript and supplemental material. In the new version of the manuscript, we have highlighted the sections of the text which have been edited.

REVIEWER 1

1. According to page 2, when in the finite range of the magnetic field near the center of the state where filling factor remains fixed, no quasiparticles can be created. However, the phase jump caused by the anyonic phase occurs in the same regime. How to explain this contradiction?

The statement that no quasiparticles can be created refers to the ideal case of zero disorder. However, in the case of a real device with finite disorder, some states with energy smaller than the gap will exist, and thus phase jumps associated with quasiparticles may occur. To clarify this point, we have edited the relevant sentence in the text as follows:

“In this regime no quasiparticles are created except for a small number which are brought to energies significantly smaller than the gap by disorder, and θ will evolve primarily due to the Aharonov-Bohm phase.”

2. I think it's better to introduce the structure of device (which one is the QPC and gate in FIG.1) and measurement setup more detail before present the measurement results.

In order to better introduce the device, we have expanded on the last paragraph of the introduction on Page 2: “Here we report on measurements and analysis of interference in a small gate-defined Fabry-Perot interferometer fabricated on a high-mobility GaAs/AlGaAs two-dimensional electron system (2DES) and specifically designed to investigate the impact of increased bulk-edge coupling. The device, shown in Fig. 1a, has lithographic dimensions

of $800 \text{ nm} \times 800 \text{ nm}$, and uses two pairs of gates forming narrow constrictions acting as QPCs to partially backscatter the incoming edge states. A pair of side gates between the two QPCs define the area of the interferometer; the QPC and side gates are highlighted in yellow in Fig. 1a. There is a center gate which is kept at ground and does not affect the 2DES density. The conductance across the device is measured using standard voltage-biased lock-in amplifier techniques.”

3. The connection between higher-order interference term and anyonic statistics is not well explained in page 3, which makes the conclusion seem abrupt.

To better explain this point, we have added the following sentence on page 3: “When the bulk is compressible, an increase of magnetic flux by Φ_0 will result in a change in quasiparticle number of -1, and if the quasiparticles at $\nu = 1/3$ are anyons, this will also result in a jump in phase of $-2\pi/3$, yielding the higher order Φ_0 periodic contributions to interference.”

4. At $\nu = 1/3$, the spacing between nodes ΔV_{SD} at finite-bias current measurement is used to calculate KI, why not use the data from differential conductance, since theory of DC current in the ref [25] doesn't include bulk-edge coupling? And in page 6, the ratio of the inner to the outer node spacing 0.85 is larger than $2/3$, not smaller.

Because the $\nu = 1/3$ state is expected to have a different finite-bias response due to Luttinger-liquid effects, we believe it is important to rely on the theoretical analysis which takes these effects into account. The theory of Ref. [25] specifically deals with total DC current rather than the differential current, so we have used the DC current to extract K_I to be consistent with this, although based in differentiating the expression for DC current, the spacing of the outer nodes in the differential conductance should give similar results (which is consistent with our observation of very similar outer node spacing in DC current and differential conductance).

We do not know if there should be corrections to Ref. [25] from bulk-edge coupling, though we leave open this possibility. Since the charge in the bulk should not change significantly when applying finite bias, it seems likely that such corrections would not be large. Therefore, we believe reasonable estimates of K_I can be obtained from our approach.

5. The article lacks commas in some places. In page 2, when introduce the filling factor, n is not well defined. It's better to unify the position of the square index like $\frac{K_I}{2}(\delta n_I)^2 + \frac{K_I}{2}(\delta n_L)^2$ in Eqn.2. In page 5, single-particle energy should be $E_{sp} = e \frac{\delta n_I^2}{2} \Delta V_{SD}$, which lack the "e" in the main text. In FIG. 5, it's better to unify the form of the axis and

legend for 5a and 5c. In page 7, according to Eqn.7, $\frac{\theta}{2\pi} = \frac{\bar{\Delta}\theta}{2\pi} + \frac{1}{3} \frac{K_{IL}}{K_L}$, not $\frac{\theta}{2\pi} = -\frac{\bar{\Delta}\theta}{2\pi} - \frac{1}{3} \frac{K_{IL}}{K_L}$.

We have added commas in several places where they are appropriate.

We have added the definitions of the n page 2 where filling factor is defined.

We have fixed Eqn. 2 to put the squared index in the correct place.

We have fixed $E_{sp} = e \frac{\delta n^2}{2} \Delta V_{SD}$.

We have set the axes to be the same for Fig. 5a and 5c.

The reviewer is correct that there was a typo in the equation for $\Delta\theta$. At the suggestion of reviewer 3, we have changed the convention so that $\Delta\theta$ refers to the change in phase for removal of a quasiparticle. Then, we have fixed the error so that $\frac{\theta_a}{2\pi} = -\frac{\bar{\Delta}\theta}{2\pi} + \frac{1}{3} \frac{K_{IL}}{K_L}$.

REVIEWER 3

1) The most important questions are whether or not the observed discrete phase jumps are indeed due to anyonic phase and the observed slightly reduced magnitude of the phase jumps are due to bulk-edge coupling, as discussed by the authors in the last section. I believe that the authors' analysis and interpretations are correct, but complications arising from the sign conventions make it rather troublesome to confirm this. For example, the authors observe phase jumps of $\Delta\theta/2\pi = -0.24$, with the negative sign explained as due to the fact that quasiparticle number decreases upon increasing magnetic field. However, eq. (20) of ref. 28, which is given for the case of quasiparticle number changing by -1, shows that the phase jump (normalized by 2π) varies from $2/3$ for the weak coupling limit ($K_{IL}/K_I = 0$) to 1 for the strong coupling limit ($K_{IL}/K_I = 1$). Since the phase is modulo 2π , this is equivalent to varying from $-1/3$ (weak-coupling limit) to 0 (strong-coupling limit), consistent with the authors' observation. But, in this comparison, the negative sign is just a matter of choice in the domain of definition, rather than being due to the fact that the quasiparticles are being removed. Perhaps the confusion occurs because the authors first introduce eq. (7) in the form for the case of adding a quasiparticle and then try to use it for comparison with experiments where they define the phase jump for removing quasiparticles. I suggest the authors always use equations for removing quasiparticles (or decreasing the net quasiparticle number) as in ref. 28 and state that they define the phase in the range from $-\pi$ to π .

We have followed the reviewers suggestion of changing the convention so that $\Delta\theta$ corresponds to a change in quasiparticle number by -1, and stated that the phase is defined from

$-\pi$ to $+\pi$. This should make comparison with experimental observations more clear.

2) On page 7, right column, there must be something wrong with the equation, $\theta_a/2\pi = -\bar{\Delta}\theta/2\pi - (1/3)K_{IL}L/K_I$. If we plug in $\bar{\Delta}\theta/2\pi = -0.24$ and $K_{IL}/K_I = 0.24$, we have $\theta_a/2\pi = -(-0.24) - (1/3) * 0.24 = 0.16$ in stead of 0.32. While removing the first minus sign on the right-hand side gives a correct result (0.32), the minus sign in the second term is inconsistent with eq. (7). It is not clear whether this is due to the difference between adding and removing a quasiparticle.

This was indeed a mistake in the equation; it should have read $\frac{\theta_a}{2\pi} = -\frac{\bar{\Delta}\theta}{2\pi} + \frac{1}{3}\frac{K_{IL}}{K_I}$. We have fixed this mistake in the manuscript.

3) For integer QH states the phase jumps associated with a change of N_L by -1 equals K_{IL}/K_I (eq. 19 of ref. 28). As the authors note at the end of page 7, the bulk-edge coupling can cause discrete phase jumps even for electrons and, if so, the ratio K_{IL}/K_I would be extracted directly from the phase jumps, as has been done in Ref. 42. However, in the $\nu = 1$ data shown in Fig.3, such discrete phase jumps are not observed, even when localized electrons or holes are introduced at lower and higher fields as manifested by the changes in the magnetic-field period. (Instead, we see modulation on the stripes, indicating contribution from other Fourier components.) I would like the authors to elaborate a bit more on the behavior of integer systems in their sample in this respect, including the $\nu = 3$ data in Fig. 4 (where features like phase jumps are seen).

While they are not easy to see without a zoomed-in view of the data, discrete jumps in phase actually are visible in our data at $\nu = 1$ in some ranges of magnetic field in the compressible regions at high and low fields. We have mentioned this in the main text, and in the supplementary material we have extended the analysis of discrete jumps in phase we used at $\nu = 1/3$ to extract the values of the phase jumps at $\nu = 1$. The average values of the phase jumps are in reasonable agreement with the values of $\frac{K_{IL}}{K_I}$ determined from the other measurements. We have added this statement to the main text discussing this: “An additional effect of bulk-edge coupling in the integer quantum Hall regime is discrete jumps in phase with when the localized electron number changes [29, 43, 48]; at $\nu = 1$ in our data jumps occur consistent with this in some ranges of magnetic field (see Supp. Section 7 and Supp. Fig 8), and similar effects can be seen at $\nu = 3$ in Fig. 4b where bulk-edge coupling is strong.”

4) The authors consistently use the value of $K_{IL} = 72\mu\text{eV}$ derived from the Coulomb

diamonds at zero field for $\nu = 3$ as well as for $\nu = 1$ and $1/3$. As the $\nu = 3$ region is surrounded by a relatively wide $\nu = 2$ incompressible strip, its island size could be smaller than for the case of $\nu = 1$ and $1/3$. Could this affect the value of K_{IL} and the authors' discussions on the $\nu = 3$ data?

This is a good point; it is certainly possible that the area encircled by the innermost edge state at $\nu = 3$ is significantly smaller, leading to greater coupling to the bulk and greater K_{IL} , leading to a larger ratio of $\frac{K_{IL}}{K_I}$ (on top of the effect of smaller K_I). We have added the following discussion of this possibility to the main text: “Additionally, because this is an inner edge mode, it may enclose an area smaller than the overall interferometer area, and thus might have greater bulk-edge coupling and K_{IL} than estimated from zero-field Coulomb blockade. This would drive the device further into the Coulomb-dominated regime.”

5) At three places in the manuscript, the authors use the verb “obscure” to describe the impact the bulk-edge coupling has on the anyonic phase. I doubt whether “obscure” is the right word because it sounds as if bulk-edge coupling only makes the anyonic phase difficult to observe but does not affect the magnitude of the phase jump.

To make the meaning of these statements more clear, we have changed the word “obscured” to “reduced” in these three instances.

Minor or technical points: - page 1 Φ_0 in the abstract is not defined.

We have added a definition for Φ_0 on page 1.

- page 1, left column The sentence in the introduction, “Additionally, evidence of non-Abelian braiding statistics has been seen in interferometry experiments at $\nu = 5/2$,” might be misleading. It sounds as if the non-Abelian braiding statistics of the $5/2$ state have already been experimentally demonstrated.

We have edited this sentence as follows to avoid confusion:

‘Additionally, interferometry has been used to study the potentially non-Abelian $\nu = 5/2$ state.’

- page 1, left column About Eq. (1) and wherever relevant, the authors should mention that the charges, when expressed with an asterisk, denote charges normalized by the elementary charge.

We have added the statement “ e^* is the quasiparticle charge on the interfering edge state normalized to the elementary electron charge e ” to clarify this point.

- page 1, right column The sentence in the introduction, “Strong bulk-edge coupling can

result in unusual interference behavior, including a decrease in magnetic flux through the interference path when magnetic field is increased, resulting in positively sloped lines of constant phase.” The authors should mention that this is referring to a plot in the magnetic field-gate voltage plane; otherwise, the term “positively sloped” would not make sense.

We have edited this sentence to read “positively sloped lines of constant phase in the magnetic field-gate voltage plane.”

- page 2, left column The statement about the constant-filling regime, “in this regime no quasiparticles are created and θ will evolve primarily due to the Aharonov-Bohm phase” appears a bit confusing, because in the experimental data shown in Fig. 1c several phase jumps due to the creation of quasiparticles are observed in the constant-filling regime.

To clarify this point, we have edited this sentence to read, “In this regime no quasiparticles are created except for a small number which are brought to energies significantly smaller than the gap by disorder, and θ will evolve primarily due to the Aharonov-Bohm phase.”

- page 2, left column Incomplete sentence: “At $\nu = 1/3$ the behavior expected for this regime where the bulk is compressible that is upon the addition of magnetic flux Φ_0 a quasiparticle will be removed from the device (or a quasihole will be added)...”

We have fixed this typo so that the sentence reads , “At $\nu = 1/3$, the behavior expected for this regime where the bulk is compressible is that upon the addition of magnetic flux Φ_0 , a quasiparticle will be removed from the device (or a quasihole will be added).”

- page 2, left column The quantity “n” should be defined when it first appears in the equation $\nu \equiv n\Phi_0/B$. This is important, since otherwise “constant n” (page 2, right column) would not make sense.

We have added a definition for the electron density n on page 2.

- page 2, left column due Φ_0 periodic — due to Φ_0 periodic

We have fixed this typo in the text.

- page 2, left column A conjunctive is missing in the long sentence starting with “In a previous work at $\nu = 1/3$ [19] the lines of constant phase were observed to shift from a negative slope near the center to zero slope at high and low field, ...”

We have edited this sentence as follows: “In a previous work at $\nu = 1/3$ [20], the lines of constant phase were observed to shift from a negative slope near the center to zero slope at high and low field, which is consistent with a shift from constant ν with an incompressible bulk to constant n with a compressible bulk.”

- page 3, right column “with zero magnetic field frequency” — “with zero magnetic-field frequency”

We have fixed this typo in the text.

- page 6, right column “More direct comparison to [25] can be made by to measurements of oscillations in the DC current” — “More direct comparison to [25] can be made by measurements of oscillations in the DC current”

We have corrected this typo.

- page 6, right column “which is somewhat smaller than the value of 2/3 predicted by Eqn. 6” Correctly, it should read, “which is somewhat larger than the value of 2/3 predicted by Eqn. 6.”

We have corrected this typo, changing “smaller” to “larger” in this sentence.

- page 7, right column “note however that the difference in sign is accounted for the fact that ...” — “note however that the difference in sign is accounted for by the fact that ...”

This sentence has been edited because we have adopted the reviewers suggestion of taking the convention for $\Delta\theta$ to correspond to a change of N by -1. The statement now reads “here the negative sign occurs because $\Delta\theta$ is for a change in quasiparticle number by -1, and the phase is defined from $-\pi$ to $+\pi$.”

- page 9, left column A predicate verb is missing in “Optical lithography and wet etching to define the mesa.”

We have fixed this sentence to read “Optical lithography and wet etching were used to define the mesa.”

- page 9, left column “mechanically polished to to make” – “mechanically polished to make”

We have fixed this typo.

- In Fig. 4c-4f and Fig. 5 (and many of the figures in the supplementary material), the fonts of the labels on ticks, axes, and color scale bars are too small.

We have increased the font size on the plots in Fig. 4 and Fig. 5 in the main text, as well as on on the plots in Supp. Fig. 2 and Supp. Fig. 5.

- Reference 40 is already published. Rep. Prog. Phys. 84, 076501 (2021).

We have updated this citation to the full published version.

LIST OF CHANGES

- Added the following passage to the introduction: “The device, shown in Fig. 1a, uses two pairs of gates forming narrow constrictions acting as QPCs to partially backscatter the incoming edge states. A pair of side gates between the two QPCs define the area of the interferometer; the QPC and side gates are highlighted in yellow in Fig. 1a. The conductance across the device is measured using standard voltage-biased lock-in amplifier techniques. ”
- Added the following sentence on Page 3: “When the bulk is compressible, an increase of magnetic flux by Φ_0 will result in a change in quasiparticle number of -1, and if the quasiparticles at $\nu = 1/3$ are anyons, this will also result in a jump in phase of $-2\pi/3$, yielding the higher order Φ_0 periodic contributions to interference.”
- Edited the sentence on Page 2 to clarify as follows: “In this regime no quasiparticles are created except for a small number which are brought to energies significantly smaller than the gap by disorder, and θ will evolve primarily due to the Aharonov-Bohm phase.”
- Defined n on page 2.
- Fixed $E_{sp} = e\frac{\delta n_I^2}{2}\Delta V_{SD}$ on page 5.
- Set the axes to be the same for Fig. 5a and 5c.
- Changed the convention of $\Delta\theta$ to correspond to a change in quasiparticle number by -1 instead of +1.
- Fixed the equation on page 7, $\frac{\theta_a}{2\pi} = -\frac{\overline{\Delta\theta}}{2\pi} + \frac{1}{3}\frac{K_{LL}}{K_I}$.
- Changed the word “obscured” to “reduced” in reference to the effect of bulk-edge coupling on anyonic phase.
- Added Ref. [14] for observation of fractional Josephson relation as a signature of fractional charge.
- Added discussion of discrete jumps due to bulk-edge coupling in the main text, as well as Supp. Section 7 and Supp. Fig. 8.

- Fixed the typo of “somewhat smaller” to “somewhat larger” when comparing the observed node spacing ratio of 0.85 to the theoretical value of $2/3$.
- Fixed typos in the equations on page 7, where the factors of δn_I were omitted.
- Added a statement that “ e^* is the quasiparticle charge on the interfering edge state normalized to the elementary electron charge e .”
- Added the statement about positively sloped lines of constant phase referring to plots in the magnetic field-gate voltage plane: “positively sloped lines of constant phase in the magnetic field-gate voltage plane”.
- Fixed the typo on page 2 changing the “due Φ_0 ” to “due to Φ_0 .”
- Changed “with zero magnetic field frequency” to “with zero magnetic-field frequency”.
- “More direct comparison to [25] can be made by to measurements of oscillations in the DC current” changed to “More direct comparison to [25] can be made by measurements of oscillations in the DC current”.
- Fixed the typo in the methods section so that the first sentence on fabrication reads this sentence to read “Optical lithography and wet etching were used to define the mesa.”
- Deleted the repeated word “to” in the methods section.
- Edited the sentence on page 1 about non-Abelian interference to read “Additionally, interferometry has been used to study the potentially non-Abelian $\nu = 5/2$ state.”
- Increased the font size on Fig. 4, Fig. 5, Supp. Fig. 2, and Supp. Fig. 5.
- Updated the citation to Ref. 41 from the arxiv version to the now-published full version.
- Removed section headings except for Introduction and Results to comply with journal format.
- Added Author Contributions and Competing Financial Interests sections to comply with journal format.

REVIEWERS' COMMENTS

Reviewer #1 (Remarks to the Author):

The authors have addressed all my concerns. I believe this manuscript is ready for publication.

Reviewer #3 (Remarks to the Author):

I have confirmed that all the comments from the reviewers are appropriately addressed in the revised manuscript. The revised manuscript is therefore suitable for publication in Nature Communications.